# Impact of Climate Change on Cassava Yield in Nigeria: An Autoregressive Distributed Lag Bound Approach

Casmir Ndukaku Anyaegbu [1], Kingsley Ezechukwu Okpara [1,*], Wirach Taweepreda [2], David Akeju [3], Kuaanan Techato [4], Robert Ugochukwu Onyeneke [5], Saran Poshyachinda [6] and Siwatt Pongpiachan [7]

[1] Environmental Management Program, Faculty of Environmental Management (FEM), Prince of Songkla University (PSU), Hat Yai 90110, Thailand

[2] Polymer Science Program, Division of Physical Science, Faculty of Science, Prince of Songkla University (PSU), Hat Yai 90110, Thailand

[3] Department of Sociology, University of Lagos, Akoka-Yaba, Lagos 100213, Nigeria

[4] Program of Sustainable Energy Management, Faculty of Environmental Management (FEM), Prince of Songkla University (PSU), Hat Yai 90110, Thailand

[5] Department of Agriculture, Alex Ekwueme Federal University Ndufu-Alike, Ikwo 482131, Nigeria

[6] National Astronomical Research Institute of Thailand (Public Organization), 260 Moo 4, T. Donkaew A. Mae Rim, Chiang Mai 50180, Thailand

[7] NIDA Center for Research & Development of Disaster Prevention & Management, School of Social and Environmental Development, National Institute of Development Administration (NIDA), 148 Moo 3, Sereethai Road, Klong-Chan, Bangkapi, Bangkok 10240, Thailand

\* Correspondence: okparakingsleyezechukwu@rschst.edu.ng

**Abstract:** Across the globe, climate change is threatening the environment, crop yield and food security. The key to ensuring a sustainable environment, crop yield increase and food security is to identify the long-term significant impact of climate change and the means of reducing the effect. This study examined the impacts of climate change on cassava yield in Nigeria. Data were sourced from the Climate Change Knowledge Portal and the Food and Agricultural Organization of the United Nations spanning from 1990 to 2019. The impact of climate change was analyzed using Autoregressive Distributed Lag Bound approach, Error Correction Model and Augmented Dickey–Fuller and Phillips–Perron tests for stationarity test. The model was subjected to diagnostic tests such as stability tests, normality tests, serial correlation tests and heteroscedasticity tests. With the exception of temperature, the study revealed that arable land, rainfall and greenhouse gases such as $C_2O$, nitrous oxide and methane had a long-term significant impact on cassava yield in Nigeria. The study also noted that methane causes long-term significant damage to cassava yield more than any other greenhouse gas and climatic variables in the study. We recommended policies and programs that facilitate the uptake of climate-smart agriculture that centers on greenhouse gas emission reduction and on crop improvement research by breeding crop varieties that will be resilient to climate shocks.

**Keywords:** ARDL; cassava yield; climate change; co-integration; ECM; greenhouse gases

## 1. Introduction

Food security is the foundation for the overall sustainable development of any economy. Climate change is significantly threatening the sustainable development of food crops across the globe and cassava production is not left out. Climate change is characterized by increased intensity and frequency of storms, drought and flooding, changed hydrological cycles and variation in precipitation [1]. The indirect effect of climate change is inimical to agricultural production through changes on species such as pollinators, pests, disease vectors and invasive species.

Climate change has a greater impact on household welfare indicators (household income, agricultural income and daily calorie consumption per capita) and food security

in SSA. The study by Economic and Policy Analysis of Climate Change (EPIC) of FAO [1] on the impacts of climate change and weather shocks in SSA revealed that climate change has significant impact on household welfare indicators. The household welfare indicators involved are the total income, agricultural income and daily calorie consumption per capita. Hence, these welfare indicators correlate with food security. In Tanzania, increased rainfall variability correlated with a 35% decrease in total income, while increased temperature fluctuation correlated with an 11% decrease in daily calorie intake [1]. FAO also had a similar report from Malawi where an increase in temperature by 1 °C (drought shock) resulted in a 19.9% decrease in consumption per capita and a 38.7% decrease in food calorie intake. The incident followed suit in Ethiopia and Niger where fluctuations in rainfall patterns and temperature changes threatened consumption expenditure, household income and food security [1]. More so, $CO_2$ emissions have impacts on agriculture and household welfare [2]. Not just in SSA alone but in the whole world, climate change is threatening the environment, crop yield and food security.

Some scholars considered climate change impact through five major channels which include: agriculture, roads, hydropower, sea-level rise and cyclones [3]. Climate change (global warming) plays a pivotal role in a loss in agricultural productivity, sea-level rise and health effects which also extends to Gross Domestic Product (GDP) growth [4]. Research has shown that climate change (coastal flooding) leads to financial instability and homelessness among urban households [5]. It has been further revealed that the agricultural sector is most susceptible to the negative impacts of climate change, especially for maize and cassava crops as their yields decrease [3]. Increasing temperature, changing rainfall patterns and frequent or severe extreme weather events (for example, heat waves, drought, floods, cold waves and storms) are listed among the indicators of climate change [6]. Each indicator of climate change has varied hidden risks which challenge our environment [7–12].

For decades, cassava has been one of the major food crops produced and consumed in Nigeria [13] with many by-products. There has been a consistent increase in the production of cassava in terms of area cultivated and yield per hectare over the last five decades in Nigeria [13]. Nigeria takes the lead for decades now as the world's largest producer of cassava with an average output of 60,001,531 million tons and 7,737,846 ha of area harvested in 2020 [13]. "Cassava is mostly grown by low-income, smallholder farmers. It made its mark in joining the lead of the few staple crops that can be produced efficiently on a small scale, without the need for mechanization, and in marginal areas with low nutrient soils and extreme weather events such as drought" [14]. As such, cassava is hardy and any extreme weather events that affect cassava will most likely affect many other staple food crops which could lead to a food crisis in Nigeria. Thus, this informed choosing cassava for this study. This proposition has been supported by a study which predicted that climate change has less impact on cassava yield relative to maize, millet and sorghum [15]. The majority of farmers in developing countries like Nigeria are mostly dependent on rain for agriculture production, thus making farmers more susceptible to climate change extreme events [16,17].

Across the globe, climate change is threatening the environment, crop yield and food security. The key to ensuring a sustainable environment, crop yield increase and food security is by identifying the long-term significant impact of climate change and means of cushioning the effect. Thus, key knowledge gaps exist in determining the climatic variables and greenhouse gases that cost more lasting damage to the cassava yield and solutions to achieve climate action. Several scholars have studied the impacts of climate change on crops but have not attributed it specifically to climatic variables such as rainfall and temperature and greenhouse gases such as $CO_2$, $N_2O$ and $CH_4$ on cassava yield in Nigeria. Determining the individual impact of climate change variables on crop yield is a gap to fill and a step toward stimulating insight into curbing the negative impacts on crop yield vis-à-vis food security. As research [18] showed that climate change affects crops differently across different parts of the world, this work is specifically centered on identifying the impact of climate change (temperature, rainfall, $CH_4$, $CO_2$ and $H_2O$) on cassava in Nigeria.

Mitigation of greenhouse gas emissions is a requirement to decelerate global warming and achieve climate action [19].

According to The Royal Society [20], from preindustrial times till the present, "the atmospheric concentration of $CO_2$ has increased by over 40%, methane by more than 150%, and nitrous oxide by almost 20%". Climate change is evident in changing trends of temperature and precipitation and increased incidences of extreme weather events such as droughts and floods. Climate change, to a large extent, is the outcome of human activities that results in the accumulation of greenhouse gas (GHG) [21]. Greenhouse gas (GHG) emissions are a major global issue due to their effects on climate and the subsequent environmental and human impacts, particularly on agricultural productivity [22–32]. The primary greenhouse gases (GHGs) include carbon dioxide ($CO_2$), methane ($CH_4$) and nitrous oxide ($N_2O$) which are emitted into the atmosphere through different human activities [23].

The human activities that lead to the emission of GHG into the environment include but are not limited to, energy supply, manufacturing, transportation, commercial and residential buildings and waste. Conventional and non-conventional agricultural practices are among the factors that influence GHG emissions substantially [23]. FAOSTAT [13] narrows down the human activities (agricultural practices) that cause GHG emissions to crop and livestock activities and forest management, and includes land use and land-use change processes. Carbon dioxide fertilizes crops which increases crop yield. However, as $CO_2$ emissions continue to contribute to climate warming, the negative impact will offset the benefit of increasing crop yield after 10 years [33]. The impact of $CO_2$ is more complicated [33]. Kim [34] observed that the application of urea fertilizer in crop production increased cumulative $N_2O$ emissions, while conventional tillage also increased $N_2O$ relative to no-tillage.

The agricultural activities that lead to GHG emissions include "burning-crop residues (0.60%), crop residues (3%), enteric fermentation (47.05%), manure applied to soil (2.80%), manure left on pasture (12.30%), manure management (6.80%), rice cultivation (12%), savanna fires (4.90%), and synthetic fertilizers (9.70%)" [13]. Figure 1 below summarizes the average percentage share of greenhouse gas (GHG) emissions from agricultural activities from 2000 to 2019 across the globe [13]. Figure 1 shows that enteric fermentation (47.05%) claimed almost half of all the GHG emissions from agricultural activities across the globe. Enteric fermentation produces methane ($CH_4$) in the rumen of ruminant animals (such as cattle, goats, sheep, etc.) as microbial fermentation takes place during the process of digestion.

"One of the biggest challenges in the 21st century is climate change which threatens food security" [14]. This situation could be associated with low agricultural productivity occasioned by climate extreme events. The growing "precipitation intensity" observed in southeastern Nigeria, occasioned by climate change is introducing "erosion and flood of different magnitude" [35]. Erosion and flooding wash off soil nutrients and destroy soil's physical structure leading to land degradation and lower productivity. There is evidence that "climate change is already having significant negative impacts on smallholder farmers" [36] and cassava production is not left out.

This paper seeks to study some identified climatic variables (like temperature and rainfall), greenhouse gases (such as $CO_2$, $CH_4$ and $N_2O$) and a covariance variable (such as the area of land used for cassava production) that have a long-term impact on the cassava yield which will lead to developing insight on how to reduce the impact of these variables on cassava yield. The results of this study could lead to enhanced food security.

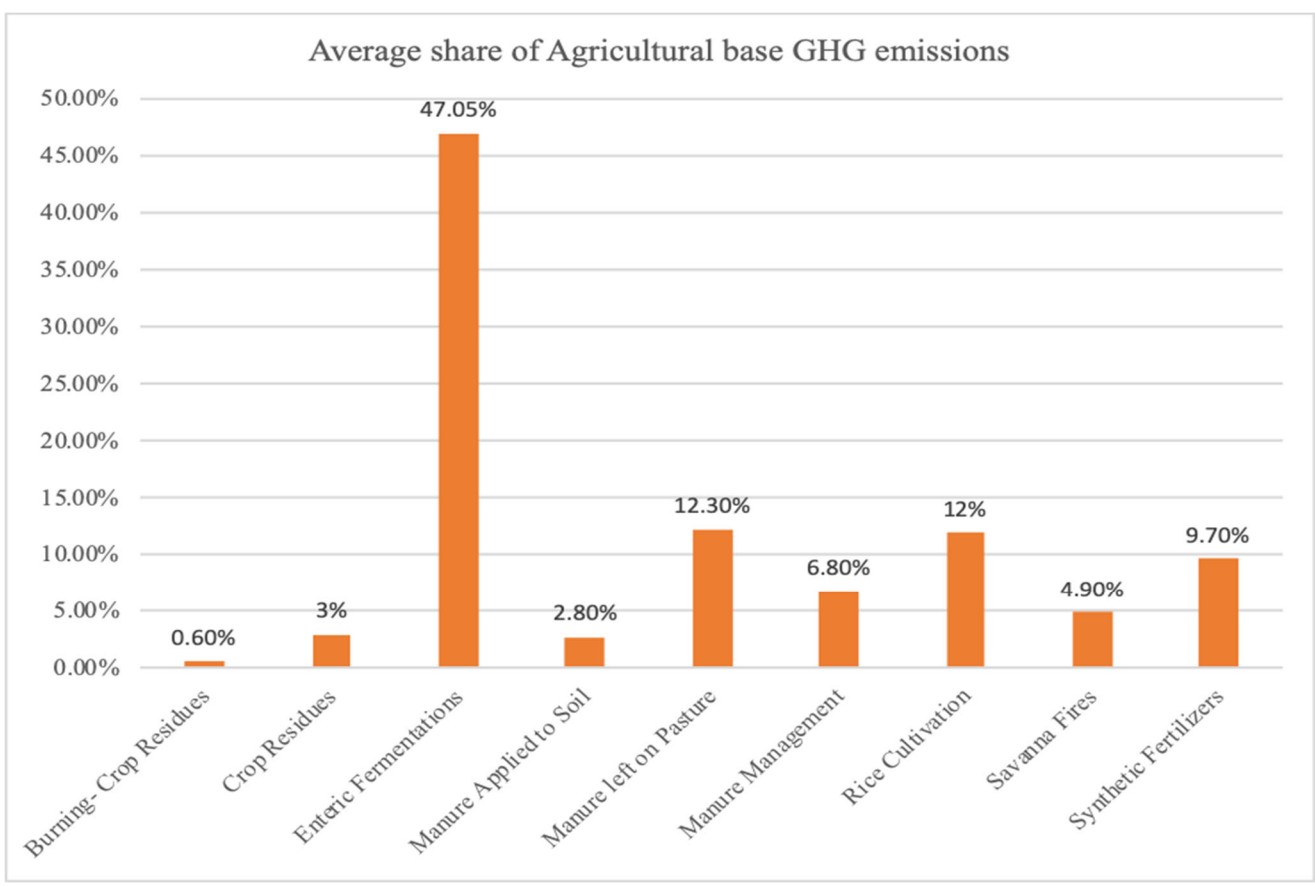

**Figure 1.** Global average share of Agro-Base GHG emissions 2000–2019 [13].

## 2. Material and Methods

*Description of Study Area*

The study was conducted in Nigeria. It is also known as the giant of Africa due to its population, landmass, human and capital development and Gross Domestic Product (GDP). Nigeria is characterized by three climatic zones which include a tropical monsoon climate in the southern part of the country with a mean annual rainfall of about 2000 mm with 31 °C and 23 °C of average day and night temperatures; a tropical savannah climate for most of the central regions with an average annual rainfall of greater than 1200 mm; 33 °C and 22 °C temperature for day and night, respectively; a Sahelian hot and semi-arid climatic condition in the northern part of Nigeria with 35 °C and 21 °C temperature for daytime and nighttime, respectively, and a 700 mm mean annual rainfall. In a chronological manner, the following are the most produced commodities in Nigeria from 1990 to 2019 which include but are not limited to cassava, yams, cereals, egg and hen in shell, vegetable primary, fruit primary, oil palm fruit, maize and sorghum [13]. It was revealed that Nigeria is leading in cassava production across the globe followed by Congo, Thailand, Brazil, Indonesia, Ghana and Angola [13].

According to the World Bank [37], Nigeria has 34 million arable land which made cultivation of cassava and other arable crops possible. In Nigeria, 19,043,008 tons and 59,411,510 tons of cassava were harvested in 1990 and 2019 in an area of land that amount to 1,634,130 ha to 7,449,387 ha, respectively [13]. The IPCC revealed that the GHG emissions in Nigeria amount to 40,222.47 of carbon dioxide ($CO_2$ eq) in 1990 and 85,256.81 of carbon dioxide ($CO_2$ eq) in 2019; 55.3 of nitrous oxide ($N_2O$) in 1990 and 103.86 of nitrous oxide ($N_2O$) in 2019; 906.04 of methane ($CH_4$) in 1990 and 2061.88 of $CH_4$ in 2019 [3].

## 3. Data Sources

The time series data used in this article are climatic variables (such as temperature and rainfall), greenhouse gases and land area in hectares across Nigeria spanning from 1990 to 2019. The climatic variables include annual mean temperature and rainfall, while greenhouse gas emission data include $CH_4$, $CO_2$ and $N_2O$. These data were obtained from the Climate Change Knowledge Portal [38] and the Food and Agricultural Organization of the United Nations [13].

## 4. Model Estimation Procedures

The long-run dynamic relationship between the yield of cassava and the predictor variables was estimated using Autoregressive Distributed Lag (ARDL) Bound approach. ARDL model was chosen for this study because it is used to determine the long-term relationship between variables under study. The relationship tends to quash when the series is integrated at order 2(1), hence showing presence of a unit root. When the series are integrated in different orders, such as 1(0) and 1(1), the Bounds Test Co-integration and Error Correction Model (ECM) of ARDL become appropriate to establish a long-run relationship in the model. The Johansen Co-integration Test is no longer valid in this study because of a combination of 1(0) and 1(1) order of integration in the series. The unit root test was to ascertain that no variable was integrated at order 2(1) and for ARDL model specification and appropriate interpretation.

The first step in the ARDL relationship analysis is the stationarity test (unit root test) [39]. Stationarity test shows the level of integration of each variable understudy. Similar work has been conducted by some scholars [39–41] using ARDL model.

The empirical application of the ARDL methodology comprises three steps:

1. identifying the order of integration of variables using the unit root tests as presented in Table 1;
2. conducting the Bounds test co-integration (long-run) relationship as presented in Table 2 and
3. estimation of an Error Correction Model (ECM) to ascertain the speed of adjustment and spurious status of the estimation.

**Table 1.** Results of the Stationarity/Unit Root test.

| Variables | At Level 1(0) | Remarks | At 1st Deference 1(1) | Remarks | *Decision: $H_0$* | Order of Integration |
|---|---|---|---|---|---|---|
| | *t*-statistic | | *t*-statistic | | | |
| Results of Augmented Dickey–Fuller Test | | | | | | |
| Y | 0.05 | Not stationary | −3.29 ** | Stationary | Reject | 1(1) at 5% |
| $X_1$ | −0.25 | Not stationary | −4.03 ** | Stationary | Reject | 1(1) at 5% |
| $X_2$ | −3.50 ** | Stationary | −4.51 ** | Stationary | Reject | 1(0) at 5% |
| $X_3$ | 1.00 | Not stationary | −4.37 ** | Stationary | Reject | 1(1) at 5% |
| $X_4$ | −0.76 | Not stationary | −4.07 ** | Stationary | Reject | 1(1) at 5% |
| $X_5$ | −0.82 | Not stationary | −4.72 ** | | Reject | 1(1) at 5% |
| $X_6$ | −2.62 | Not stationary | −13.00 ** | Stationary | Reject | 1(1) at 5% |
| Results of Phillips–Perron Test | | | | | | |
| Y | −2.35 | Not stationary | −16.67 *** | Stationary | Reject | 1(1) at 1% |
| $X_1$ | −1.78 | Not stationary | −7.67 *** | Stationary | Reject | 1(1) at 1% |
| $X_2$ | −3.46 ** | Stationary | 19.18 *** | Stationary | Reject | 1(0) at 1% |
| $X_3$ | 0.10 | Not stationary | −4.35 *** | Stationary | Reject | 1(1) at 1% |
| $X_4$ | −0.63 | Not stationary | −6.51 *** | Stationary | Reject | 1(1) at 1% |
| $X_5$ | −0.89 | Not stationary | −4.61 *** | Stationary | Reject | 1(1) at 1% |
| $X_6$ | −6.58 | Not stationary | −15.11 *** | Stationary | Reject | 1(1) at 1% |

Note: ** and *** represent significance levels at 5% and 1%, respectively. $H_0$ = series have a unit root. Data source: Output from Eviews 12.

**Table 2.** Bounds test for co-integration.

| Equation | F-Statistic | Lower Bound 1(0) 5% | Upper Bound 1(1) 5% |
|---|---|---|---|
| lnYield = lnLand lnTemp lnCO$_2$ lnN$_2$O lnCH$_4$ lnRF | 9.27 | 2.45 | 3.61 |

Data source: Output from Eviews 12.

### 4.1. Unit Root Test

A unit root test analysis of each of the time series variables under the study was conducted to determine the order of integration. Thus, the Augmented Dickey–Fuller (ADF) and the Phillips–Perron (PP) tests were employed to determine the order of integration of the chosen variables.

The model for the ADF test is presented in Equation (1):

$$\Delta Y_t = \alpha + \partial Y_{t-1} + \Sigma \gamma \Delta Y_{t-j} + e_t \tag{1}$$

where,

$Y = series\ of\ tested$
$\Delta Yt = first\ difference\ of\ Yt$
$\partial = $ test difference coefficient
$j = $ lag length chosen for ADF
$e_t = $ white noise
$t = $ time or trend variable

Here, the significance $\partial$ was tested against the null hypothesis (H$_o$), $\partial = 0$ and alternative hypothesis, $\partial < 0$. If we do not reject the null, the series is non-stationary. Hence, if the hypothesis of non-stationarity cannot be rejected, each of the variables was differenced until they became stationary (that is significant at the 5% level). At this point, the existence of a unit root was rejected. The stationarity test is also known as the unit root test. The unit root test was to ascertain that no variable was integrated at order 2 and for ARDL model specification [42]. Thereafter, co-integration was conducted.

### 4.2. Co-Integration Analysis: ARDL Bounds Test

The Bounds test ARDL model was adopted in testing the long-run relationship (co-integration) between the cassava yield and predictor variables. This has been adopted by some scholars in similar work [43–45]. The predictor variables are arable land in hectares, climatic variables (temperature and rainfall) and greenhouse gases like CO$_2$, N$_2$O and CH$_4$. As soon as co-integration is established, the conditional ARDL (p, q1, q2, q3, q4, q5 and q6) and the long-run model for Y$_t$ can be specified as:

$$lnY_t = \beta_0 + \sum_{i=1}^{p} \alpha_1 \ln Y_{t-1} + \sum_{i=0}^{q1} \alpha_2 lnland_{1t-1} + \sum_{i=0}^{q2} \alpha_3 lntemp_{2t-1} + \\ \sum_{i=0}^{q3} \alpha_4 \ln CO_{23t-1} + \sum_{i=0}^{q4} \alpha_5 lnN_2O_{4t-1} + \sum_{i=0}^{q5} \alpha_6 lnCH_{45t-1} + \sum_{i=0}^{q6} \alpha_7 lnRF_{6t-1} \tag{2}$$

This involves selecting the orders of the ARDL (p, q1, q2, q3, q4, q5 and q6) model in the six predictor variables using Akaike Information Criterion as cited in a literature [46].

### 4.3. Error Correction Model (ECM)

The ECM is specified in Equation (3):

$$\Delta Y_t = \alpha_0 + \alpha_1 \Delta \check{Z}_t - \alpha_2 (Y_t - Z_t)_{t-1} + e_t \tag{3}$$

where,

$\check{Z}_t = the\ vector\ of\ explanatory\ variables$
$Y_t\ and\ Z_t = the\ co-integrating\ variables$

$\alpha_2 = error\ correction\ term\ (ECT)$
$e_t = error\ term$

In line with some scholars [46], the ECM is specified as:

$$\Delta Y_t = \beta_0 + \sum_{i=1}^{p} \alpha_1 \Delta Y_{t-1} + \sum_{i=0}^{q1} \alpha_2 \Delta land_{1t-1} + \sum_{i=0}^{q2} \alpha_3 \Delta temp_{2t-1} +$$
$$\sum_{i=0}^{q3} \alpha_4 \Delta CO_{23t-1} + \sum_{i=0}^{q4} \alpha_5 \Delta N_2 O_{4t-1} + \sum_{i=0}^{q5} \alpha_6 \Delta CH_{45t-1} + \sum_{i=0}^{q6} \alpha_{7\Delta} RF_{6t-1} -$$
$$\alpha(Y - land - temp - CO_2 - N_2O - CH_4 - RF) + e_t$$

$$(4)$$

## 5. Results and Discussion

### 5.1. ADF Test for Stationarity (Unit Root Test)

This research work studied the long-run implication of climate change on cassava yield in Nigeria. Arable land available for cultivation, climatic variables and greenhouse gases were regressed against cassava yield. We used the Augmented Dickey–Fuller (ADF) test and the Phillips–Perron (PP) test to determine the stationarity of the dependent and independent variables. Table 1 shows the summary statistics of the stationarity test and the variables under consideration. The variables include: cassava yield, area of land cassava was cultivated on, mean temperature, carbon iv oxide ($CO_2$), nitrous oxide ($N_2O$), methane ($CH_4$) and rainfall (RF). These were converted to a natural log for ease of analysis.

The ADF test of *t*-statistics for cassava yield (Y), area of land cultivated ($X_1$), temperature ($X_2$), $CO_2$ ($X_3$), $N_2O$ ($X_4$), $CH_4$ ($X_5$) and rainfall ($X_6$) were $-3.29$, $-4.03$, $-3.50$, $-4.37$, $-4.07$, $-4.72$ and $-13.00$, respectively, and statistically significant at 5% level. In the same order, PP test of *t*-statistics were $-16.67$, $-7.67$, $-3.46$, $-4.35$, $-6.51$, $-4.61$ and $-15.1$, respectively, as presented in Table 1.

In this study, the ADF test and PP test results at level and at first difference were in the discourse of the unit root test. The results show the presence of co-integration between the dependent and independent variables which informed the decision to conduct the Bounds test for co-integration. Hence, the results of the test show that only temperature was stationary at the level with an order of integration of 1(0), while cassava yield, area of land cultivated, $CO_2$, $N_2O$, $CH_4$ and rainfall were stationary at the first difference with an order of integration of 1(1), respectively. The study revealed that the variables under review influenced cassava production over time. Thus, some scholars [47–49] had similar findings in Nigeria. Specifically, Onyeneke [18] further revealed that the majority of climatic variables on crop production show stationarity at their first differencing 1(1) and others at level 1(0). This is in line with the results of this study under review.

There is a combination of 1(0) and 1(1) order of integration in the series. The series are integrated of different orders, hence Bounds test Co-integration and ECM of ARDL is appropriate to establish a long-run relationship in the model. The Johansen Co-integration Test is no longer valid in this study because of a combination of 1(0) and 1(1) order of integration in the series. The unit root test was to ascertain that no variable was integrated at order 2(1) and for ARDL model specification.

### 5.2. Bounds Test for Co-Integration

Table 2 presents *F*-statistic, lower bound and upper bound for long-run relationship. The *F*-statistic is 9.27 which is significantly higher than both the lower bound of 2.45 and the upper bound of 3.61 at a 5% level. This implies that there is a long-run term relationship between dependent and independent variables in the model. Table 2 shows that the null hypothesis of no co-integration is rejected at a 5% level of significance which confirms that there is co-integration among the variables.

### 5.3. Long-Run Impacts of Climate Change on Cassava Yield

The long-run estimates of the cassava yield presented the area of land used for cassava cultivation, temperature, $CO_2$, $N_2O$, $CH_4$ and rainfall as regressors of the estimate in Table 3 while cassava yield is the dependent variable. Out of six regressors (predictor variables), five were found to be significant at a 10% level while only one was not significant. This

implies that the five significant predictor variables have a long-term relationship with cassava yield (dependent variable). The significant variables are the area of land used for cassava production, $CO_2$, $N_2O$, $CH_4$ and rainfall while the temperature was not significant.

**Table 3.** Long-run (LR) estimate of cassava yield.

| Predictor Variables | Coefficient | Standard Error | *t*-Statistic | *p*-Value |
|---|---|---|---|---|
| Area of land (ha) | 0.2 | 0.11 | 1.81 | 0.09 * |
| Temperature | 0.02 | 0.03 | 0.62 | 0.54 |
| $CO_2$ | 3.24 | 1.66 | 1.95 | 0.07 * |
| $N_2O$ | 0.82 | 0.43 | 1.92 | 0.07 * |
| $CH_4$ | −0.46 | 0.26 | −1.75 | 0.10 * |
| Rainfall | 0.07 | 0.04 | 1.80 | 0.09 * |
| Constant | 26.24 | 16.85 | −1.56 | 0.14 |

The independent variables are significant at a 10% (*) level. Data source: Output from Eviews 12.

Table 3 presents the long-run implications of climate change on cassava yield in Nigeria. Here, cassava yield was regressed against arable land, temperature, rainfall and greenhouse gases such as $CO_2$, $N_2O$ and $CH_4$.

In Table 3, the study revealed that the area of land in hectares has a positive coefficient estimate of 0.20 at a 10% level of significance. The result shows that a unit increase in the area of land for cassava production will increase cassava yield by 20%. This shows that arable land has a long-term relationship with cassava yield. This implies that the large expanse of arable land available for cultivation is a comparative advantage for cassava farmers in Nigeria to increase their yield in the long run. Land has been a limiting factor in crop production, especially in the southern part of Nigeria which is the research area. The land tenure system practice in Nigeria especially in the southeast part of the country has made access to land difficult. By implication, the higher the arable land available for cassava production, the more the economies of scale and the higher the cassava yield. Thus, the availability of arable land for cultivation has a positive long-term significant impact on the cassava yield in Nigeria.

In this study, carbon iv oxide ($CO_2$) has a positive coefficient estimate of 0.07 at a 10% level of significance. This shows that the higher the $CO_2$, the higher the cassava yield. $CO_2$ is essential during the process of photosynthesis. Photosynthesis is the process by which crops such as cassava create sugar and oxygen using sunlight, water ($H_2O$) and carbon dioxide ($CO_2$). The sugar in the form of glucose is utilized during the cellular respiration of the cassava crop. More so, glucose is used by the cassava crop as an important source of carbon to produce cassava stems and roots thereby increasing the yield. However, increased $CO_2$ above the optimum requirement of crops in the presence of warmer temperatures and wetter climates promotes weed, pest and fungi infestation. Hence, the result establishes that $CO_2$ emissions have a positive long-term significant impact on the cassava yield in Nigeria. This is in contrast with the findings of some scholars, according to Schipani [50]. He reported that $CO_2$ shows positive impact on crops but will diminish and have a negative impact when atmospheric $CO_2$ reaches a saturation point.

The study revealed that rainfall has a positive coefficient estimate of 0.07 at a 10% level of significance. This indicates that a unit increase in rainfall within the optimum requirement of cassava will lead to a 7% increase in cassava yield. In Sub-Saharan Africa (SSA), the majority of farmers depend on rain for their crop production, thus making farmers more susceptible to climate change extreme events [16,17]. As such, changes in rainfall affect crop production. More so, changes in climatic factors such as rainfall and temperature exert significant influences on the mean yield levels and yield variance of pulses [51]. Cultivation of cassava requires water in the form of rainfall for photosynthesis which promotes the performance of cassava. Thus, the higher the rain within the optimum requirement of cassava, the higher the yield of cassava. Hence, rainfall has a positive long-term significant impact on the cassava yield in Nigeria. However, the growing "precipitation intensity" observed in southeastern Nigeria, occasioned by climate change is

introducing "erosion and flood of different magnitude" [35]. Erosion and flooding wash off soil nutrients and destroy soil's physical structure leading to land degradation and lower productivity.

Nitrous oxide ($N_2O$) has a positive coefficient estimate of 0.82 and a significant impact on cassava yield at a 10% level. By implication, a unit increase in the supply of $N_2O$ will lead to an 82% increase in cassava production. This indicates that the higher the $N_2O$ application, the higher the cassava yield. Hence, $N_2O$ has a long-term positive effect on cassava yield. Nitrogen is an essential nutrient for plant growth, which is why it is used in the production of synthetic fertilizers applied to farmland across the globe [52]. However, the emission of greenhouse gases like $N_2O$ contributes to climate change which in turn affects crop production negatively. Researchers have noted that the anthropogenic source of $N_2O$ has jumped by 30% and that the majority of that increase (87%) was stimulated by agriculture [52].

Methane ($CH_4$) has a negative coefficient estimate of $-0.46$ at a 10% level of significance. By implication, a unit increase in $CH_4$ supply to cassava will result to a 46% decrease in the yield of cassava in Nigeria. This indicates that an increase in the supply of $CH_4$ will significantly lower the cassava yield in Nigeria. The result reveals that methane has a long-term negative impact on cassava yield in Nigeria. Methane is a greenhouse gas (GHG) emission which contributes to global warming. The study revealed that methane caused long-term significant damage to cassava more than any other greenhouse gas in the study. This is in line with some scholars [53] who assumed maize losses to be $CH_4$ and wheat losses to be $CO_2$ (due to warming) and $CH_4$. The study revealed that other GHG emissions like $CO_2$ and $N_2O$ show a positive significant relationship with cassava yield.

$CH_4$ emissions are very harmful to crops as the gas increases surface ozone that causes harmful chlorosis (yellowing of the leaves) [33]. This is in line with the result of this study. Thus, $CH_4$ emissions promote global warming, cost damage to public health and lessen the yield and productivity of agricultural and forest ecosystems [54].

Livestock produces significant amounts of methane ($CH_4$) as part of their normal digestive processes. Methane-reducing feed additives and supplements inhibit methanogens in the rumen and later reduce enteric $CH_4$ emissions. This will help to achieve Sustainable Development Goal 13 (climate action) by 2030. Methane-reducing feed additives and supplements include: (1) synthetic chemicals, (2) natural supplements and compounds, such as tannins and seaweed and (3) fats and oils. Some researchers [55] revealed that the burning of biomass and landfills involving waste releases greenhouse gases like $CO_2$ and $CH_4$ into the atmosphere. By implication, combustion efficiency of biomass and better waste disposal other than landfills will reduce GHG emissions. While a study [56] noted that the reduction potential of GHG emissions depends on (1) number of crops per year, (2) residue management, (3) the amount of applied irrigation water and (4) sand content.

The result showed that temperature had a positive coefficient estimate of 0.02 but was not significant. The result suggests that a unit rise in temperature will not significantly raise cassava output in Nigeria. Climate change causes extreme temperature and rainfall leading to floods and droughts which harm crops and reduces yield. Temperature above optimal requirement of cassava crop tends to be harsh on the crop and reduces vegetation performance and yield.

*5.4. Error Correction Model (ECM) Regression*

Table 4 presents the Error Correction Model (ECM) estimation.

Table 4 was used to interpret the Error Correction Model (ECM) and spurious nature of the model. The coefficient of Error Correction Term ($-0.57$) is negative and statistically significant at a 1% level. The negative and significant coefficient estimate of ECM indicates that there is a co-integrating relationship between cassava yield and its determinants.

**Table 4.** Error Correction Model (ECM) Estimation.

| Variables | Coefficient | Standard Error | *t*-Stat | *p*-Value |
|---|---|---|---|---|
| CointEq (−1) | −0.57 | 0.06 | −9.24 | 0.00 |
| Constant | −26.24 | 2.84 | −9.23 | 0.00 |
| $R^2$ | 0.80 | - | - | - |
| Adjusted $R^2$ | 0.77 | - | - | - |
| F-statistic | 33.03 | - | - | - |
| Prob(F-statistic) | 0.00 | - | - | - |
| Durbin-Watson stat | 2.50 | - | - | - |

Data source: Output from Eviews 12.

In Table 4, the magnitude of the coefficient estimate of ECM suggests that 57% of the disequilibrium caused by previous years' shocks converges back to the long-run equilibrium in the current year. This reveals that the speed of adjustment is above average and it is a bit fast to adjust to the long-term equilibrium. By implication, the independent variables (land area, temperature, $CO_2$, $N_2O$, $CH_4$ and rainfall) would adjust any negative short-run and long-run shocks to the cassava production in the long run. Some scholars [18,42,47–49] have shown similar findings in their research.

Table 4 also presents the spurious nature of the model. Here, the Durbin–Watson statistic (2.50) is greater than the R-squared ($R^2$) value of 0.80 which implies that the model is not spurious.

*5.5. Diagnostic Test*

Figure 2 presents the CUSUM test of stability. The rule of thumb in the CUSUM test is that if the blue line at the center did not touch the upper and lower bound lines, the model is stable and if otherwise, the model is not stable as presented in Figure 2. Table 5 presents the diagnostic test assumptions of the ARDL model of the estimate. These diagnostic test assumptions include stability test, normality test, serial correlation test and heteroscedasticity test.

**Table 5.** Diagnostic test.

| Diagnostic Test | Test | Probability Value | *t*-Statistic | F-Statistic | Prob Chi-Square |
|---|---|---|---|---|---|
| Stability test | Ramsey RESET test | - | 0.79 | 0.79 | - |
| Normality test | Jarque–Bera stat | 0.13 | - | - | - |
| Serial correlation test | LM test | 0.17 | - | - | 0.09 |
| Heteroscedasticity test | Breusch–Pagan–Godfrey | 0.46 | - | - | 0.39 |

In this study, Ramsey RESET and CUSUM tests were adopted in the stability test of the ARDL model. The rule of thumb for Ramsey RESET test is that if the *t*-statistic and *F*-statistic are greater than 0.05 (5%), it shows that the model is stable. The result revealed a *t*-statistic of 0.79 and an *F*-statistic of 0.79 which is greater than 0.05. Thus, the model is stable. To corroborate the stability test, the CUSUM test was also conducted which further proved that the model is stable.

The normality test was conducted using the Jarque–Bera statistic. Here, the probability value (0.13) is greater than 0.05 (5%), which shows that the residue of the model is normally distributed.

Data source: Output from Eviews 12. A serial correlation test was also conducted employing the LM test. Here, the probability value of 0.17 and the probability of chi-square value of 0.09 were greater than 0.05 (5%). This implies that there is no serial correlation in the model.

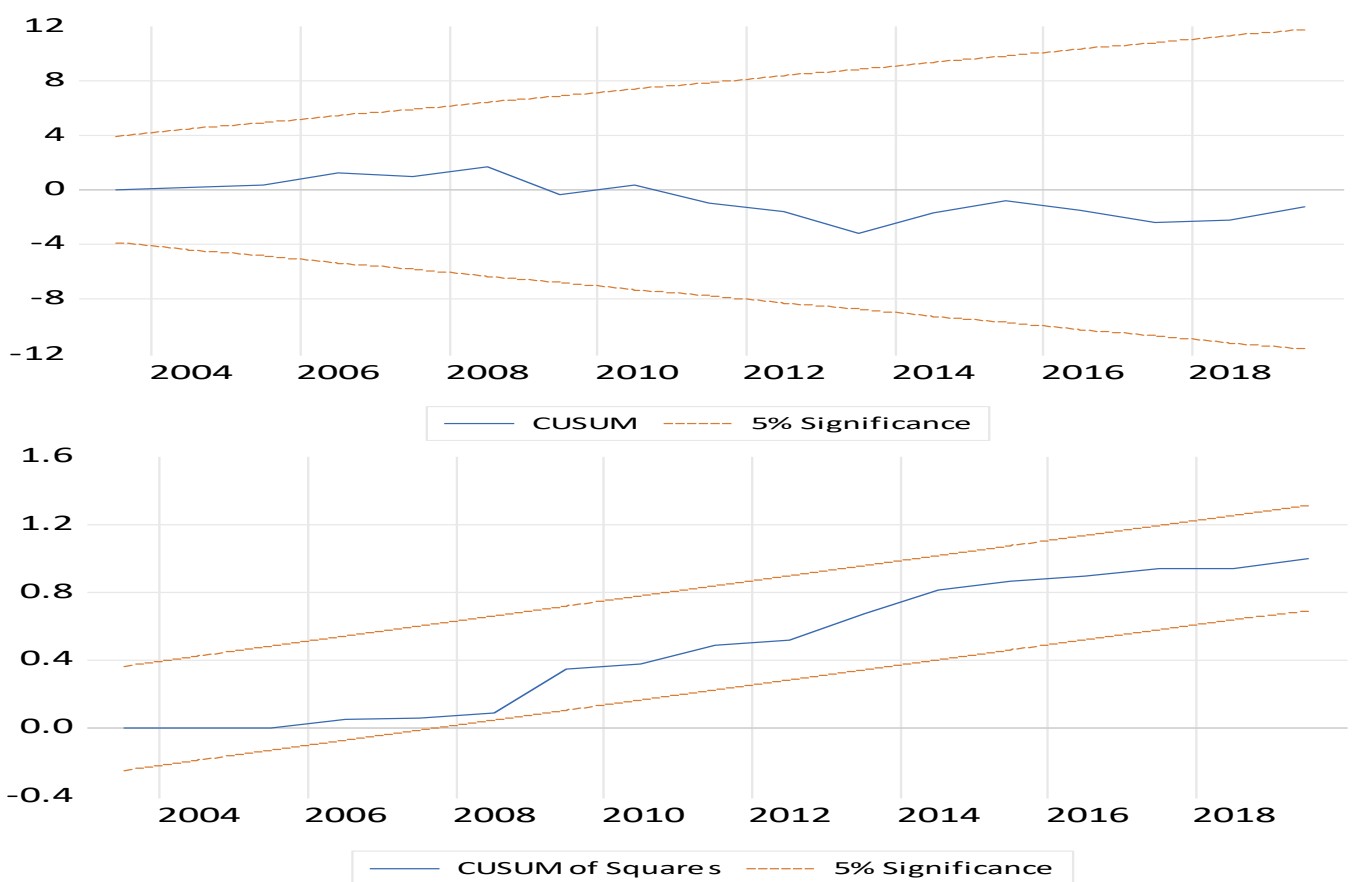

**Figure 2.** CUSUM stability test.

The heteroscedasticity test was conducted using the Breusch–Pagan–Godfrey test. Here, the probability value of 0.46 and probability of chi-square value of 0.39 were greater than 0.05 (5%). This shows that the model is homoscedasticity. Hence, the diagnostic test revealed that the ARDL model used for the study is a good fit and is reliable. The results showed the appropriateness of ARDL to model the impact of climatic variables, greenhouse gases and land on cassava production in Nigeria.

## 6. Conclusions

The study examined the long-run implication of climate change and greenhouse gases on cassava yield in Nigeria. Arable land available for cultivation, climatic variables and greenhouse gases were regressed against cassava yield. Specifically, the regressors are arable land (ha) used for cassava cultivation, temperature, $CO_2$, $N_2O$, $CH_4$ and rainfall. The stationarity test (unit root test) was conducted using the ADF and PP tests which show that all the variables are significant at 5% and 1% levels, respectively. The unit root test results show that only temperature showed stationarity at level with an order of integration of 1(0), while cassava yield, area of land cultivated, $CO_2$, $N_2O$, $CH_4$ and rainfall were stationary at first difference with an order of the integration of 1(1).

The series are integrated at different orders, hence Bounds test Co-integration and Error Correction Model of ARDL were adjudged appropriate to establish a long-run relationship between the cassava yield and the predictor variables (regressors). It implies that a unit increase in the supply of $CH_4$ will lead to a 46% decrease in cassava yield in Nigeria. The result of the Bounds test for co-integration rejected the null hypothesis of no co-integration at a 5% level of significance which established that there is a co-integration among the variables under consideration. The study revealed that arable land, climatic variables and greenhouse gases had a long-term significant impact on the cassava yield in Nigeria. However, the temperature is not significant. The study also revealed that methane

caused a long-term significant damage to cassava more than any other greenhouse gas and climatic variable in the study.

*Recommendations*

The study recommends climate-smart agriculture (CSA) and manure management as a low-cost countermeasure with a high reduction potential of GHG emissions across the globe to aid in achieving SDGs 2 and 13 by 2030, as CSA was developed in 2010 by FAO to achieve three main objectives: (1) sustainably increasing agricultural productivity and incomes; (2) adapting and building resilience to climate change and (3) reduction and/or removal of greenhouse gas emissions, where possible.

**Author Contributions:** Conceptualization, C.N.A., K.E.O., W.T., D.A., K.T., R.U.O., S.P. (Saran Poshyachinda) and S.P. (Siwatt Pongpiachan); Methodology, C.N.A., K.T. and S.P. (Siwatt Pongpiachan); Software, C.N.A.; Validation, W.T., S.P. (Saran Poshyachinda) and S.P. (Siwatt Pongpiachan); Formal analysis, C.N.A., K.E.O., K.T. and R.U.O.; Investigation, C.N.A. and D.A.; Resources, C.N.A. and D.A.; Data curation, C.N.A.; Writing—original draft, C.N.A.; Writing—review & editing, C.N.A., K.E.O., W.T., D.A., K.T., R.U.O., S.P. (Saran Poshyachinda) and S.P. (Siwatt Pongpiachan); Visualization, C.N.A. and S.P. (Saran Poshyachinda); Supervision, K.E.O., W.T., D.A., K.T. and S.P. (Saran Poshyachinda). All authors have read and agreed to the published version of the manuscript.

**Funding:** The APC was funded by Prof. Siwatt Pongpiachan of National Astronomical Research Institute of Thailand.

**Institutional Review Board Statement:** Not applicable.

**Informed Consent Statement:** Informed consent was obtained from all subjects involved in the study.

**Data Availability Statement:** All data used in this paper are publicly available.

**Acknowledgments:** This research was an excerpt of an M.Sc. dissertation of the first author. The authors would like to acknowledge Prince of Songkla University, Ministry of Higher Education, Science, Research and Innovation for the tuition-free program and research assistance. All individuals included in this section have consented to the acknowledgement.

**Conflicts of Interest:** The authors declared that they have no known conflict of interest.

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
