# Peer review of "Impact of Climate Change on Cassava Yield in Nigeria: An Autoregressive Distributed Lag Bound Approach"

_agriculture, doi:10.3390/agriculture13010080_

Round 1

Reviewer 1 Report

Dear Editor

Major corrections needed

Based on the article received, I feel that the manuscript could be reconsidered for publication after considering all the major revisions attached below:

1.     Please revise this sentence:

“Data obtained were analyzed using Autoregressive Distributed Lag Bound approach, error correction model and Augmented Dickey Fuller test for stationarity test”

2.     Please revise this section based on your key findings and relevant policy implications:

“Climate-smart agriculture, manure management in crops and methane-reducing feed additives and supplements in livestock rearing are recommended to reduce greenhouse gases to aid in achieving SDGs 2 and 13 by 2030.”

3.     The policy implications part is missing in the abstract. 

4.     In the Introduction section, please clarify the justification for this research. The introduction section is too long and failed to show the research gap and novelty of this research. The problem statement is not well organized and not clear to the reader. There is no methodological explanation or reason for choosing logistic regression. Also, there is a considerable gap between the novelty of research and the discussion of existing studies. Please make it as elaborate as you can. Substantial changes and revisions are required in the introduction section. Please make a story and try to find out the importance of this research.

5.     Please revise this sentence: “The study was conducted in Nigeria which is the most populous black nation. It is not good to use the term “black”.

6.     Please revise all equations format.

7.     Please add Phillips–Perron (PP) test for stationarity.

8.     Please add the stability test graphical results in the appendix section.

9.     Please check the font size of your entire manuscript.

Finally, I would like to say that the research topic is important in the present context. Research ideas are also good, but you should focus on the consistency of your writing.

Author Response

Dear reviewer,

The responses are made below.

  1. Please revise this sentence:

“Data obtained were analyzed using Autoregressive Distributed Lag Bound approach, error correction model and Augmented Dickey-Fuller test for stationarity test”

Response:

The impact of climate change was analyzed using Autoregressive Distributed Lag Bound approach, error correction model and Augmented Dickey-Fuller and Phillips Perrons tests for stationarity test.

  1. Please revise this section based on your key findings and relevant policy implications:

“Climate-smart agriculture, manure management in crops and methane-reducing feed additives and supplements in livestock rearing are recommended to reduce greenhouse gases to aid in achieving SDGs 2 and 13 by 2030.”

Response:

We recommended policies and programs that facilitate the uptake of climate-smart agriculture that centres on greenhouse gas emission reduction and on crop improvement research by breeding crop varieties that will be resilient to climate shocks.

  1. The policy implications part is missing in the abstract. 

Response:

We recommended policies and programs that facilitate the uptake of climate-smart agriculture that centres on greenhouse gas emission reduction and on crop improvement research by breeding crop varieties that will be resilient to climate shocks.

  1. In the Introduction section, please clarify the justification for this research. The introduction section is too long and failed to show the research gap and novelty of this research. The problem statement is not well organized and not clear to the reader. There is no methodological explanation or reason for choosing logistic regression. Also, there is a considerable gap between the novelty of research and the discussion of existing studies. Please make it as elaborate as you can. Substantial changes and revisions are required in the introduction section. Please make a story and try to find out the importance of this research.

Response:

We have reduced the introduction section.

Justification, research gap and novelty of this research were captured in these two paragraphs and other paragraphs

For decades, cassava has been one of the major food crops produced and consumed in Nigeria [13] with many by-products. There has been a consistent increase in the production of cassava in terms of area cultivated and yield per hectare over the last 5 decades in Nigeria [13]. Nigeria take the lead for decades now as the world’s largest producer of cassava with an average output of 60,001,531 million tonnes and 7,737,846 ha of area harvested in 2020 [13]. “Cassava is mostly grown by low-income, smallholder farmers. It made its mark in joining the lead of the few staple crops that can be produced efficiently on a small scale, without the need for mechanization, and in marginal areas with low nutrient soils and extreme weather events such as drought” [13]. As such, cassava is hardy and any extreme weather events that affect cassava will most likely affect many other staple food crops which could lead to a food crisis in Nigeria. Thus, this informed choosing cassava for this study. This proposition has been supported by a study which predicted that climate change has less impact on cassava yield relative to maize, millet and sorghum [15]. The majority of farmers in developing countries like Nigeria are mostly dependent on rain-fed for agriculture production, thus making farmers more susceptible to climate change extreme events [16, 17].  

Across the globe, climate change is threatening the environment, crop yield and food security. The key to ensuring a sustainable environment, and crop yield increase and food security is by identifying the long-term significant impact of climate change and means of cushioning the effect. Thus, key knowledge gaps exist in determining the climatic variables and greenhouse gases that cost more lasting damage to the cassava yield and solutions to achieve climate action. Several scholars have studied the impacts of climate change on crops but have not attributed it specifically to climatic variables such as rainfall and temperature and greenhouse gases such as CO2, N2O and CH4 on cassava yield in Nigeria. Determining the individual impact of climate change variables on crop yield is a gap to fill and a step toward stimulating insight into curbing the negative impacts on crop yield vis-a-vis food security. As research [18]  showed that climate change affects crops differently across different parts of the world, this work is specifically centered on identifying the impact of climate change (temperature, rainfall, CH4, CO2 and H2O) on cassava in Nigeria. Mitigation of greenhouse gas emissions is a requirement to decelerate global warming and achieve climate action [19].  

Below is part of the methodological explanations

The long-run dynamic relationship between the yield of cassava and the predictor variables was estimated using Autoregressive Distributed Lag (ARDL) Bound approach. ARDL model was chosen for this study because it is used to determine the long-term relationship between variables under study. The relationship tends to quash when the series is integrated at order 2(1), hence showing presence of a unit root. When the series are integrated in different orders, such as 1(0) and 1(1), the Bounds Test Co-integration and Error Correction Model (ECM) of ARDL become appropriate to establish a long-run relationship in the model. The Johansen Co-integration Test is no longer valid in this study because of a combination of 1(0) and 1(1) order of integration in the series. The unit root test was to ascertain that no variable was integrated at order two 2(1) and for ARDL model specification and appropriate interpretation.

The first step in the ARDL relationship analysis is the stationarity test (unit root test) [40]. Stationarity test shows the level of integration of each variable understudy. Similar work has been done by some scholars [40-42] using ARDL model.

The empirical application of the ARDL methodology comprises three steps:

  1. identifying the order of integration of variables using the unit root tests as presented in Table 1.
  2. conducting the bounds test co-integration (long-run) relationship as presented in Table 2; and
  3. estimation of an Error Correction Model (ECM) to ascertain the speed of adjustment and spurious status of the estimation as presented in table 4.

Logistic regression was not used in this study, rather we used Autoregressive Distributed Lag (ARDL) Bound Approach

  1. Please revise this sentence: “The study was conducted in Nigeria which is the most populous black nation. It is not good to use the term “black”.

Response:

The study was conducted in Nigeria.

  1. Please revise all equations format.

Response:

The equations has been re-formatted

  1. Please add Phillips–Perron (PP) test for stationarity.

Response:

Phillips-Perron (PP) test has been added in table 1. But it is not for stability. It is used to test stationarity/unit root of the model. It is an alternative test to Augmented Dickey-Fuller (ADF) test. In this case, it complemented the ADF test.

  1. Please add the stability test graphical results in the appendix section.

Response:

The graphical result of stability test (CUSUM test) has been added as figure 2.

  1. Please check the font size of your entire manuscript.

Response:

The font size of the entire manuscript has been made uniform except for the tables and the title of the paper.

The attached file is the revised manuscript.

Accept my esteemed regards.

Reviewer 2 Report

The manuscript needs major revisions in all sections.

see the attached pdf file

data given is very less for such type of study

needs good figures of data to improve paper quality

too much generalizations which already reported

discussion id very bad, there is no support of literature in discussion sections

both conclusions and recommendations are very weak, only significant words are reported but not supported by tables/figures etc

Author Response

Dear Reviewer,

The corrections observed have been effected in the manuscript. 

The generalisations observed have been removed. 

The discussions have been improved with the support of literature.

The data for this study span from 1990 to 2019 which is 30 years interval. The authors believed that the data were adequate for this study. 

2020 to 2021 data were not available in the repositories at the time of this research.

You requested for percentage increase or the degree of impact.

Response:

The Autoregressive Distributed Lag (ARDL) Bound approach was used to model the impact of climate change on cassava yield. ARDL is used to determine whether there is a long-term relationship between the cassava yield and the independent variables under study. The result is used to ascertain the impact of the variables on the cassava yield. The model does not present the degrees of impact.  But the coefficient of the estimate is used to predict the percentage/degree of impact if there is a unit increase in the predictor variables.

Of course, these predictions were made in the discussion section.

The attached file is the revised manuscript. 

Accept my esteemed regards

Reviewer 3 Report

This is an important study reported by the authors to identify the regulatory points of cassava yield in Nigeria using models and statistical approaches. Such studies provide the long term effect of climate change on the yield of crops of a particular region. Authors have studied temperature/rainfall factors and greenhouse gases which are likely to affect the yield in that particular region. The manuscript is suitable for inclusion in the Agriculture. However, following minor changes are required before acceptance of the manuscript.

- Introduction is too sketchy. It needs to be succinct to emphasize the problem and purpose of the study. Introduction should focus on why the study is on cassava. For this some description related to cassava should be shifted to Introduction.

- Authors should see if a comparative table to show the effects of temperature and rainfall, and green house gases be included so that a quick results can be seen on the degree of effectiveness on cassava yield.

- The references of FAO and IPCC should also be from their recent reports. These reports are published and available.

- place comma before and after ‘respectively’ throughout the manuscript.

- symbol of degree Celsius should be addressed in the manuscript.

- Recommendation made should be reduced to adjust the exact findings of the study.

Author Response

Dear reviewer,

Below are the comments and their responses:

  • Introduction is too sketchy. It needs to be succinct to emphasize the problem and purpose of the study. Introduction should focus on why the study is on cassava. For this some description related to cassava should be shifted to Introduction.

Responses:

For decades, cassava has been one of the major food crops produced and consumed in Nigeria [13] with many by-products. There has been a consistent increase in the production of cassava in terms of area cultivated and yield per hectare over the last 5 decades in Nigeria [13]. Nigeria take the lead for decades now as the world’s largest producer of cassava with an average output of 60,001,531 million tonnes and 7,737,846 ha of area harvested in 2020 [13]. “Cassava is mostly grown by low-income, smallholder farmers. It made its mark in joining the lead of the few staple crops that can be produced efficiently on a small scale, without the need for mechanization, and in marginal areas with low nutrient soils and extreme weather events such as drought” [13]. As such, cassava is hardy and any extreme weather events that affect cassava will most likely affect many other staple food crops which could lead to a food crisis in Nigeria. Thus, this informed choosing cassava for this study. This proposition has been supported by a study which predicted that climate change has less impact on cassava yield relative to maize, millet and sorghum [15]. The majority of farmers in developing countries like Nigeria are mostly dependent on rain-fed for agriculture production, thus making farmers more susceptible to climate change extreme events [16, 17].  

Across the globe, climate change is threatening the environment, crop yield and food security. The key to ensuring a sustainable environment, and crop yield increase and food security is by identifying the long-term significant impact of climate change and means of cushioning the effect. Thus, key knowledge gaps exist in determining the climatic variables and greenhouse gases that cost more lasting damage to the cassava yield and solutions to achieve climate action. Several scholars have studied the impacts of climate change on crops but have not attributed it specifically to climatic variables such as rainfall and temperature and greenhouse gases such as CO2, N2O and CH4 on cassava yield in Nigeria. Determining the individual impact of climate change variables on crop yield is a gap to fill and a step toward stimulating insight into curbing the negative impacts on crop yield vis-a-vis food security. As research [18]  showed that climate change affects crops differently across different parts of the world, this work is specifically centered on identifying the impact of climate change (temperature, rainfall, CH4, CO2 and H2O) on cassava in Nigeria. Mitigation of greenhouse gas emissions is a requirement to decelerate global warming and achieve climate action [19].  

- Authors should see if a comparative table to show the effects of temperature and rainfall, and green house gases be included so that a quick results can be seen on the degree of effectiveness on cassava yield.

Response:

The authors believed that table 3 in the manuscript was a kind of comparative table. The table presents the variables and each of their coefficient of estimates was used to predict the future impact of these variables on the cassava yield when there is a change in the individual variable.

This table was also used to compare the variable that has impact on cassava yield.

- The references of FAO and IPCC should also be from their recent reports. These reports are published and available.

The references have been updated

- place comma before and after ‘respectively’ throughout the manuscript.

Response: This has been addressed

- symbol of degree Celsius should be addressed in the manuscript.

Response: This has been addressed

- Recommendation made should be reduced to adjust the exact findings of the study

Response:

We recommended policies and programs that facilitate the uptake of climate-smart agriculture that centres on greenhouse gas emission reduction and on crop improvement research by breeding crop varieties that will be resilient to climate shocks.

The attached file is the revised manuscript. 

Accept my esteemed regards

Round 2

Reviewer 1 Report

Thanks for your hard work. Congratulations.

Reviewer 2 Report

The manuscript is revised according to the comments and I will suggest its publication now. However, better the authors check it again for minor errors or English if any.